# Adaptive Learning Framework (Alef) in UAE Public Schools from the Parents' Perspective

**Nazera Emara [1], Nagla Ali [2] and Othman Abu Khurma [2],***

1   Department of Professional Learning Communities (PLC), Emirates Schools Establishment, Al Qusais, Dubai 8589, United Arab Emirates
2   Department of Curriculum and Instruction, Emirates College for Advanced Education, Abu Dhabi 126662, United Arab Emirates
*   Correspondence: othman.abukhurma@ecae.ac.ae

**Abstract:** This study aimed to evaluate the Adaptive Learning Framework (Alef) platform and determine how parents perceive the implementation of such a program to support their children's learning. Alef is a smart learning program that is mandated in United Arab Emirates (UAE) public schools to promote student-centered, interactive, and differentiated learning and personalized experiences. The participants were parents of students enrolled in grades 9–12 in UAE public high schools. The study used semi-structured interviews to gather qualitative data to delve deeply into parents' perceptions of how Alef supports their children's learning. The results showed that Alef offers stimulating and engaging educational experiences and encourages independent learning. It was found that students' learning was bolstered when using Alef. In addition, the results indicated the existence of some challenges stemming from the usage of Alef that could have an impact on students' learning and motivation and, ultimately, the sustainability of such a program. Recommendations are provided to overcome these challenges.

**Keywords:** e-learning; Adaptive Learning Framework (Alef); artificial intelligence; parents' perception; personalized learning

## 1. Introduction

Educational systems are facing new challenges as a result of the Fourth Industrial Revolution (4IR, IR 4.0) (Yuhastina et al. 2020; Koizumi 2019). A considerable technological upheaval has been brought about by the 4IR, including the introduction of AI, the Internet of Things (IoT), cloud computing, digital games, etc. (Ahmad et al. 2022; Liu and Stephen 2019). Educational systems are under intense pressure from the technological revolution, which demands graduates who can use new technologies to tackle critical societal issues in innovative and crucial ways (Alhamuddin et al. 2022). Educational systems are now being challenged to prepare students for jobs that do not yet exist, technologies that have not been invented, and problems that have not been encountered (World Economic Forum 2018). One way to overcome this challenge is through the integration of 4IR technologies into teaching and learning. Technology-based pedagogical practices should be implemented in schools to promote students' motivation and independence and provide tailored and convenient access to information (Khurma et al. 2023; Ally and Wark 2020).

Artificial intelligence (AI) is one technology that has been rapidly advanced by the 4IR. The term "artificial intelligence", coined by McCarthy in 1956, is associated with the science and engineering of making intelligent machines. The study of AI was founded on the premise that the central property of human intelligence can be mimicked and simulated by a machine (Lu 2019; Luxton 2016). According to Ilkka (2018), although AI has been around for several decades, the recent industrial revolution has accelerated the urgent need for the implementation of AI in many fields, including education. He postulates that AI and machine learning will have implications for workforce preparation and teaching and

learning practices (Koizumi 2019; Lee et al. 2018). With the rise of AI, education systems around the world face two challenges: exploiting AI to improve the educational process and preparing students with new skills for increasingly automated economies and societies (Vincent-Lancrin and van der Vlies 2020).

Recognizing the above challenges and the importance of artificial intelligence in our future society, the UAE government and its leaders announced an AI strategy in October 2017. This strategy was created to help achieve Centennial 2071 objectives by enhancing the performance of many governmental sectors and investing in AI and aimed to develop AI applications in many areas, including education. The objectives of the AI strategy are to build the UAE's reputation as an AI destination, attract and train talents for future jobs enabled by AI, provide world-leading research capabilities to work with industries, and up-skill and re-skill talents (UAE Centennial 2071 2018; UAE Government Leaders Programme 2018; UAE 2031 2018; Halaweh 2018). In response, the UAE Ministry of Education (MoE) implemented a program called the Adaptive Learning Framework (Alef) in its public high schools. AI is at the heart of Alef, an interactive shareware program that tailors the content provided to each student based on their individual needs, input, and identified strengths and weaknesses. The Alef program is available for all core subjects and is aligned with the MoE curriculum. It provides students with immediate feedback and gives educators and parents access to analytical data and information (Alkhalisi 2019; Alyammahi 2019).

A small body of research has revealed that the Alef program has positive implications for students' learning. In one study, it was found that the immediate feedback and person-alized instruction provided by this system not only helped students in re-evaluating their progress but also helped teachers in re-evaluating their students' grasp of the concepts and adjusting their teaching accordingly (Alyammahi 2019). Teachers' reports of the Alef program's usefulness in tracking their students' progress were cited by Almohammadi et al. (2017). Moreover, Alef gives parents the ability to oversee their children's academic performance and development (Cetinkaya 2017).

Although the Alef program has been in place since 2015, few studies have been conducted to investigate its role, its impact on students' learning, and how students and teachers perceive the role it plays in improving students' learning. In addition, not a single study has been conducted to investigate the perspectives of parents regarding the role of the Alef program in supporting their children's learning. Since students use the Alef program from their homes, parental engagement, input, and perspective are crucial. Therefore, this study aimed to investigate the role that the Alef program plays in supporting students from the perspective of their parents and investigate the challenges faced by parents and their children when using this program.

## 2. Literature Review

The UAE's Ministry of Education introduced Alef to all public high schools to fulfill the national agenda of the UAE and provide its educational system with the most recent innovations in digital education, such as artificial intelligence and big data (Al Arood and Aljallad 2020). This will help the UAE further its national aim of creating a world-class educational system with the highest possible standards. Alef supports the pedagogical integration of technology into teaching and learning, as it is one of the most significant axes of change in the educational process and one of the most prominent pillars of a twenty-first-century human renaissance. By implementing Alef, teachers and parents are able to evaluate students' performance, subject comprehension, and time spent on each session in real-time, which enables them to identify each student's strengths and weaknesses through the intelligent framework used to construct Alef (Abdallah 2018; Media and News 2020).

The intelligent framework of Alef is comparable to the framework proposed by Faisal et al. (2015). Their model blends crowdsourcing into the creation and presentation of instructional materials, thereby enabling intelligent e-learning systems to provide a mean-ingful, individualized, and pedagogically sound learning experience. The framework comprises learner, course, teacher, and recommender systems. The learner model is created

and kept in a database with comprehensive information about the learner's profile and learning preferences. The framework gives crowdsourcing feedback indicating whether or not students have mastered the subject matter. This feedback is utilized to monitor each student's strengths and weaknesses and improve their preparation for the next lesson. The course model incorporates course prerequisites, goals, topics, and course materials. Except for course materials, the repository stores all information while the recommender system dynamically modifies it. The teacher model consists of an expert system modified via crowdsourcing and a knowledge base that incorporates the knowledge and experiences of many teachers in the same field. The recommender system suggests pedagogically sound content and methods for learning. The resulting recommendations improve the learner profile, the learning environment, and the learner's preferences. The role of learning management systems in managing student learning and providing feedback was also confirmed by Adzharuddin and Ling (2013).

The aforementioned framework gives Alef a number of advantages. One of these advantages is the adoption of a student-centered teaching approach and the empowerment of teachers to encourage independent learning and boost students' engagement (Chiu and Chai 2020). According to Clark and Kaw (2020), students using adaptive learning software are exposed to a variety of learning formats (such as videos, text, graphics, or simulations), assessments, and immediate feedback as they move along a personalized learning path. This path may be one where mastery is being demonstrated or one where remediation is being offered. Similarly, Serrano et al. (2019) confirmed the effective role of technology-based learning platforms in enhancing student engagement and improving learning outcomes. The ability of Alef to encourage differentiation in learning, as well as independent learning, is another advantage of using the program. Because of the instant feedback they receive about their students' progress, teachers can focus their efforts on differentiating learning to different groups of students (Eaton et al. 2018). Most of these advantages were evident in Alyammahi's (2019) study, which was implemented in one public school in the UAE. The results of the study demonstrated that the use of Alef increased students' interest in and engagement with core subjects. All of the students who participated in the study reported that using Alef motivated them to learn and made learning enjoyable. The Alef program also helped students become more self-reliant and confident in their academic abilities. Additionally, it assisted the students in grasping key concepts, which, in turn, boosted their academic achievement. The findings of a study by Alhosani (2019) confirmed that Alef provided teachers with live data that enabled them to distinguish between students who were having difficulty and those who were performing well, thereby supporting them. Alahmari and Kyei-Blankson (2016) also confirmed the role of an e-learning system in promoting teacher collaboration with students and parents for enhanced learning outcomes. In addition, other findings demonstrated that students' engagement and motivation to learn increased as a result of using this individualized learning system, which offered students engaging digital instructional content (De Oliveira et al. 2019).

The opportunity provided by Alef to follow students' progress is not limited only to the students and their teachers. Alef provides parents with an application that allows them to participate in their children's learning journey (Alhosani 2019). Multiple studies have found that when parents are involved in their children's education, their academic performance improves. One of the most positive outcomes of parents' engagement is the promotion of self-regulatory skills (Jeynes 2012). Parental engagement refers to the engagement of parents in activities that advance the education and learning of their children in order to increase their academic and social well-being (Novianti and Garzia 2020; Fan and Williams 2010; Lui et al. 2020). In addition to achievement and academic performance, parents' engagement in the education and learning of their children is associated with improved self-regulation, stronger study habits, and more positive attitudes toward learning (Xu and Wu 2013; Obradović et al. (2021). The use of interactive technologies with accessible resources can help in developing the engagement of parents in their children's learning

(Jayawardena et al. 2020; Ewin et al. 2021). This enables data-driven educational interactions between teachers, students, and parents (Starčič and Vukan 2019).

Alef offers an entirely new communication framework between students, teachers, parents, and technology. According to Kent et al. (2022), this framework establishes a collaboration paradigm between parents and teachers. Alef provides teachers with monitoring tools that enable them to "concoct" and track the progress of individual students (Bieliková et al. 2014). There are two potential benefits for the teacher: (a) they can allocate their valuable time to providing one-on-one assistance where it is most needed, and (b) they can be provided with a summary of common difficulties on a topic that may require a whole-class intervention while also identifying individual difficulties (Alnusairat 2022). The purpose of detecting and showing this type of information is to enhance students' self-regulated learning skills. The program offers parents the same chances as teachers to contribute to the progress of their children. Discussions between children and their parents regarding their progress empower both sides since everyone can genuinely share their own experiences.

In Alef's performance screen, parents are able to monitor their children's progress in every lesson in each subject as well as the number of lessons and the details about each lesson. In addition, a message assessing the student's level is displayed. The report screen allows parents to view a weekly report of their child's performance. It also shows the topics that the child excelled in during the week, the topics that need development, and the activities that the child must perform to develop certain skills (Home Alef Education 2021; Pawlak 2019). Parents receive immediate real-time feedback about their child's progress. In addition, parents are informed about their children's learning gaps, which will enable parents to provide them with the appropriate support (Alkhalisi 2019).

Despite all the advantages that Alef provides for students' learning, Abujaja and Abukari (2019) identified two major challenges associated with Alef. First, some teachers rely excessively on Alef to deliver their lessons, which limits students' opportunities to engage in real-world and hands-on activities and experiences. Second, in some Alef lessons, students can bypass the video explanation and go straight to the assessment, where the students choose random answers to determine the correct answer. Then, the students go through a second attempt to choose the correct answer. If this happens, the data presented to the teachers about their students' learning would mislead the teachers and make it harder for them to make the right decisions related to their students' learning. Alyammahi (2019) identified the need for cultural transformation and change management to enhance teachers and students' adaptation to the system since teachers and students reported they found it overwhelming to adapt to the new system and identified it as a major challenge. Similarly, Bieliková et al. (2014) noted that various types of learning objects (explanations, exercises, questions) are not effectively distinguished in the platform, affecting the recommendations for outside material made by the system. Thus, a high volume of recommendations for outside material is presented, increasing the load on students for learning. Although independent learning has been praised in the literature, Mirata and Bergamin (2019) noted that adaptive learning frameworks put more of a burden on students by making them solve questions on their own, which might be difficult for them. The monotonous and fixed interface of the learning software was also found to be a key challenge in managing the software, as revealed by Gee (2022) and Kumar et al. (2020). Alahmari and Kyei-Blankson (2016) also identified some challenges related to e-learning systems, including teacher development and training and internet access.

## 3. Methodology

The current study used a qualitative approach to answer the research questions. This approach was used because it was most suitable for examining the phenomena under study and trying to comprehend and interpret them (Basit 2003). Here, we attempted to understand parents' perspectives on the role of Alef in supporting their children's learning. In addition, we investigated the challenges faced by both children and their parents when

using Alef. Semi-structured interviews were chosen to collect data regarding parents' perspectives. The semi-structured interviews were considered the most effective method of data collection in this study for gaining a deeper understanding of parents' perspectives. The semi-structured interview consisted of a sequence of prop-assisted questions posed to the participants (Vasileiou et al. 2018).

### 3.1. Study Participants

The target population of this study was parents (the father or the mother) of children studying in one of the public high schools in Abu Dhabi. The following criteria were applied to select parents for this study: (a) the parent should have a child who is in grades 9–12 in a public high school in Abu Dhabi; (b) the parent and the child should have used Alef for at least one year; and (c) the child should use Alef to learn all the core subjects (math, science, English, Arabic, and Islam). Twenty-one parents voluntarily agreed to participate in the study out of thirty-five parents who met the above criteria and were invited to participate.

### 3.2. Data Collection Tools

For data collection, we conducted semi-structured interviews with the participants. Semi-structured interviews allow participants to speak candidly about issues that are personally meaningful and crucial to them (Horton et al. 2004; Newcomer et al. 2015). During semi-structured interviews, a trustworthy and collaborative atmosphere must be established for the participants to be able to give their opinions without any fear. Conversely, the interview also permits psychological, social, and educational inquiry (Braun and Clarke 2019). The semi-structured interviews comprised seven interview questions that inquired about the participants' perception of the Alef program in supporting their children's learning. Participants were recruited via an email invitation to parents who met the inclusion criteria. The email included an introduction and details regarding the study. Consent forms were then explained to and signed by the parents who volunteered to take part in the study. The interviews with the parents lasted roughly thirty minutes. Before beginning the interview, participants were reminded about the consent form and the recording of the interview. All the interviews were conducted in Arabic, the parents' native language.

The following interview questions were used to collect data from the parents:

a. How long has your child used the Alef platform? How often does your child use the Alef platform?
b. Please explain your experience and your child's experience with the Alef platform.
c. What data does Alef provide you related to your child's progress? Please provide some examples. How do you use/act on these data?
d. In which ways do you think the Alef platform supports your child's learning? Please provide some examples.
e. Have you noticed any differences in your child's learning, motivation, achievement, or skills after using the Alef platform?
f. What recommendations have you heard from your child related to the use of the Alef platform? What are your suggestions to solve the problem/issues faced by your child?
g. What suggestions would you like to add to the Alef platform to improve and develop the platform to better support your child's learning?
h. Would you encourage your child's access to and use of the Alef platform in upcoming years? Why?

### 3.3. Data Analysis

The data from the semi-structured interviews were analyzed manually. The analysis of these qualitative data was performed manually due to the fact that it is a creative process of inductive reasoning, thinking, and theorizing; interviews are qualitative tools that can

be analyzed by converting the answers into code and then themes (Kallio et al. 2016). The individual interviews were analyzed using a thematic approach, which is a method used in the analysis of qualitative data whereby the data are classified and placed into specific categories. Thematic analyses systematically explain the mechanisms of coding and analyzing qualitative data, which can then be linked to broader theories or concepts (Braun and Clarke 2019). All the interviews were transcribed to a Word document and analyzed. After rereading the interviews, a list of codes was created. The qualitative results are presented as a narrative.

## 4. Results

To answer the research questions, an investigation into how parents felt about the implementation of the Alef program was conducted through semi-structured individual interviews with the parents based on the indicators of the Adaptive Learning Framework (Alef) in UAE public schools. Each theme that evolved from the qualitative data analysis is presented separately. Then, the section concludes by presenting the parents' recommendations of ways in which Alef could be improved to better support their children's learning.

### 4.1. Learning Support

Parents voiced that they would keep encouraging their children to use Alef at home because Alef supports their children's learning. Alef provides different resources to students, including activities, exercises, different types of assessments, and questions that help in supporting their learning and improving their achievement levels and scores. The main theme that appeared in the parents' responses was the accessibility of Alef from home, which supported the idea that Alef supports their children's learning.

*My child has been using Alef on regular basis to solve activities related to the subjects. The program is very useful in reviewing science and mathematics concepts.*

*Learning stations help my daughter to review the scientific material. The assessment tests also contribute to determining her level of comprehension and retrying in case of not achieve the desired score.*

*I encourage my daughter to continue using the Alef platform because it is in constant development and helps raise her academic level.*

*A wonderful experience because it makes the parent aware of the curriculum and continues their learning at home.*

*I encourage my son to use Alef because it supports learning at home.*

*Alef is supporting my daughter in competing for the work that was not accomplished during the lessons.*

*Lots of exercises, reviews after each unit and different questions that support my daughter's learning at home.*

*I will keep encouraging my son to use Alef to develop fundamental skills.*

*The strategies are all-inclusive, covering every facet of the educational experience. Alef's lessons present the learner with objectives, games, and assignments; they then instantly correct their work and give the student feedback on their progress.*

### 4.2. Independent Learning

The idea that the Alef program promotes independent learning was cited multiple times in the literature as a significant way in which Alef helps students further their education. This support is provided and experienced in a variety of different ways, such as the questions that are presented by the program after each session, the self-based videos, the learning stations, etc. Alef provides students with the option to prepare for the following session by flipping the traditional classroom lesson format into a more student-centered

learning environment. The students prepare for hands-on class activities that will be based on the work that they have completed at home by watching videos and answering questions about what they have seen.

> *Alef guarantees self-education to students due to the presence of videos explaining the lesson in addition to the variety of questions.*

> *Alef encourages independent learning by providing lessons and videos that may be viewed in the comfort of one's own home.*

> *My daughter relies on Alef platform for self-learning the themes when she prepares for the lesson before class.*

> *Because of Alef my son developed independent learning and self-learning skills.*

> *Alef's comprehensive nature is apparent in its many instructional videos and leveled questions, the latter of which allow me as a parent to gauge my daughter's proficiency by seeing whether or not she can decipher questions at a beginner, intermediate, or advanced level. If she can answer the advanced questions, it means she has grasped the material and is doing an excellent job of learning it.*

> *The major idea station and accompanying examples contribute to Alef's comprehensiveness.*

### 4.3. Motivation

The parents mentioned that their children were more motivated to learn when they used Alef because of the built-in reward systems, such as seeing the word "completed" after each lesson (which indicates that the students can move on to the next level, as in gamification), the feedback system, the presentation of the content in an appealing way, and the use of videos. One mother shared how she noticed her daughter's joy while she was working through the program with her daughter, saying, "I notice my daughter's happiness while performing her lessons in Alef". Another parent shared a perspective that was very similar to this one when she said that her child was "always motivated to utilize Alef to get as many stars as possible". Some parents stated that their children's learning improved as a direct result of using Alef regularly and that their children's motivation and accomplishment levels also increased.

> *My son has improved a lot and became motivated to learn.*

> *Training students on the pattern of exam questions and motivating them with stars. Some teachers transform these stars into weekly appreciation certificates.*

> *The number of stars and the percentage of achievement determines each student's strengths and weaknesses.*

> *The trend of the program in diversifying the ways of displaying the materials is interesting and not boring, as it is suitable for the scientific and technical development that the country is witnessing.*

> *Every participant remarked that the different components of the lessons with Alef are sequenced and logical, meaning that one activity leads to another activity. This method motivates my son to always upgrade to the next level.*

> *My daughter is quite motivated by the number of stars she earns in the program; these stars serve as a visual representation of her progress, and she is very proud of them.*

### 4.4. Monitoring Students' Progress

The opportunity that Alef provides for parents to keep track of their children's progress in school using the parent application was cited by all parents as a significant benefit to their children's learning. When the parents were asked how they felt about being able to monitor their children's progress using Alef, their responses included those shown below.

> *I receive immediate reports about my daughter's progress and performance on my phone.*

*The exit ticket that follows each lesson is a source of data that shows me as a parent the extent to which my daughter comprehended the lesson.*

*By using Alef application, I can see a report detailing my daughter's progress and the number of stars she has earned.*

*My son's progress and areas of difficulty are graphically shown in Alef.*

*After each lesson, my son's score appears when he has finished answering the questions, and the percentage gives me peace of mind as a mother because it shows me the extent to which my son understands the lesson. If he doesn't answer well, I encourage him to go through the lesson again and improve his score.*

*Provides students' progress and level of achievement in lessons.*

*Provides the results and grades for each lesson, the progress of students, and the level in each lesson.*

*It contains a program for the parents to follow up on their child's level of progress and the messages sent from the teacher to the parents contribute to the child's academic and behavioral progress.*

*4.5. Challenges Faced*

Most of the time, students use Alef from home. This allows parents to closely monitor and talk to their children as they use the program. Sixty percent of the parents who were interviewed believed that the videos embedded in the lessons did not provide a sufficient explanation covering the questions that would follow them. All the parents reported that there was a significant amount of pressure on their children as a result of trying to complete an excessive number of assignments for different subjects using Alef. The main challenge faced by the students and parents is the number of subjects and lessons that need to be completed by the students every day. To clarify, some subjects have five classes per week, which means that the number of Alef lessons that needs to be completed daily is too high. This puts students under a great deal of strain, especially on the days when they must take the monthly assessment. Below are some answers provided by parents with respect to this issue.

*My daughter has been assigned a lot of work in Alef which requires her to be on the computer for lengthy periods.*

*My daughter struggles to complete all of the assignments in her science and mathematics classes because they need higher-level thinking and more time. Although I agree that it is important to foster higher-order thinking skills, I feel that my daughter needs additional time to finish the assignments at hand without being stressed.*

*In addition to the challenges associated with the number of tasks and lessons that need to be completed every day, during the interview, 90% of parents reported that some of the questions presented after each lesson were of high difficulty and sometimes very complicated. The following are some examples of the parents' answers related to the difficulty of the questions in the program:*

*My daughter always complains about the difficulty and unclarity of the questions.*

*Science and mathematics questions are sometimes very difficult for my son.*

*A big challenge related to using Alef is that there is not enough time at home due to the long school hours and a large number of subjects.*

*When the students use Alef, they miss the opportunity to apply what is learned.*

*The inability of the student to re-solve the exit ticket is one of the most problems that my daughter complains about.*

In summary, the reported challenges associated with the adaptive Alef system are, in fact, normal challenges found with the majority of online platforms.

*4.6. Recommendations*

The parents were asked to reflect on how Alef can be improved to better support their children's learning and what is expected from the school to help improve the implementation of this program in different subject areas. The suggestions they made included establishing more connections and coordination between the subject teachers in the tasks and lessons assigned to the students on the Alef program and not totally depending on Alef for every single lesson. In addition, it was suggested that only a certain percentage of lessons in all subjects should be offered using Alef. Other suggestions were associated with logistics, such as allocating more time to complete the tasks that took place in parallel with other tasks in other subjects and, in some cases, with the monthly assessments.

> *There are many Alef lessons for all subjects, and I recommend for the school focus the use of the program on certain subjects and limited lessons.*

> *In order to prevent students from spending all day in front of the computer screen, I recommend that teachers work together to decide which lessons in which subjects will be delivered* via *Alef.*

> *I suggest reducing the number of lessons and giving more practical examples to clarify the content.*

> *To clarify some of the vocabularies, I recommend integrating the Arabic translation system to only translate the scientific terminology.*

> *The lessons in Alef should focus more on the practical implementation of the knowledge learned than the knowledge itself.*

> *Forty percent of the participants talked about the need of improving the structure of Alef interface, the parents stated that:*

> *I suggest adding an attractive visual interface using pictures and colors. The interface color should be changed from time to time, so the students do not feel bored from dealing with the same interface and colors on regular basis.*

> *Three parents talked about the need to improve the reward system and to offer the ability to convert the stars to grades that counts towards their final grade in each subject or if this is not applicable it could be converted to a physical award.*

> *I suggest allocating a day for each subject that the student uses Alef to reduce the burden at home.*

> *Adding more questions to the exit ticket, thus the calculation of the final grades is not only based on three questions. Having only three questions is very frustrating for the student because if my daughter answered one question wrong, she loses 33% of her grade.*

> *One of my suggestions for the platform is to add other types of educational and interesting games for students.*

> *Simplifying the scientific material presented and only using Alef as an e-learning program and eliminating the use of other platforms such as LMS.*

## 5. Discussion

The results of the current study showed that parents perceive the Alef program as strong support to their children's learning. Parents believe that the Alef program creates a motivational learning experience for their children by using several interactive features such as videos, assessments, adaptive learning, and various activity stations, including the big idea station that precedes entry to the lesson. The results indicated that the Alef program allows parents to easily monitor and evaluate their children's academic progress through several approaches, including but not limited to stars, graphs, charts, feedback, easy-to-read academic-level pictures, etc. The results also revealed that the Alef program supports students' learning by providing them with individualized and personalized learning experiences based on their inputs, which helps in closing any academic gaps in

their learning. These findings are supported by Alyammahi's (2019) study, which found that the use of Alef boosted students' interest and engagement in core subjects, motivated them to learn, made learning fun, and increased their independence and self-assurance in their academic abilities. Abdallah (2018) further underlined that, as one of the most important axes of change in the educational process and one of the most visible pillars of a twenty-first-century human renaissance, Alef supports the pedagogical integration of technology into teaching and learning. The intelligent framework used to build Alef allows teachers and parents to identify each student's strengths and weaknesses, enhancing their learning outcomes. Serrano et al. (2019) also stressed the role of technology-based learning platforms for enhanced student learning. It is thus argued that, by incorporating Alef, teachers and parents are able to support students' learning and evaluate their performance, subject comprehension, and time spent on each session in real-time.

The parents reported that the use of Alef supported and promoted their children's independence in learning and increased their self-regulation, a result that was also confirmed by Ally and Wark (2020) and Chiu and Chai (2020). The children of the respondents in the present study were able to identify their weaknesses and create strategies to overcome them and achieve learning outcomes. The parents believed that their children were motivated due to the adaptive learning they received based on their interests, needs, and abilities. This is corroborated by Serrano et al. (2019), who argue that technology-based learning platforms are capable of incorporating various media and activities for enhanced student engagement, leading to increased student motivation and interest. In addition, the fact that the Alef program is available online gives parents the opportunity to follow their children's progress anywhere and anytime. This is supported by Alkhalisi (2019), who insists that parents feel empowered to monitor their children's progress through learning management systems. Another aspect that parents considered a successful feature that motivated their children to learn is that Alef possesses the same features as social networks, digital gaming, etc. It can be said that the Alef program communicates with children using their preferred language and citizenship, which is technology.

The results of some studies contradict the current study's results, while others are in line with it. For example, the theme of Alef supporting students' learning is confirmed by Abdallah's (2018) study. The Alef program supports learning from the perspective of the parent due to the comprehensiveness and sequencing of the learning stations and the logical organization of activities. The current study's findings support those of Alkhalisi (2019); Eaton et al. (2018); Alyammahi (2019); Adzharuddin and Ling (2013); and Alahmari and Kyei-Blankson (2016).

Existing research asserts that Alef encourages independent or comprehensive learning (Ally and Wark 2020; Chiu and Chai 2020). Here, the parents confirmed that the Alef platform substantially encourages student autonomy and self-education. In addition, the platform's material is marked by its thoroughness because it offers videos, activities, and tests with student performance feedback for every session. This is confirmed by Clark and Kaw (2020), who note the possibility of using various media and activities in an adaptive learning platform. The findings of enhanced student motivation and engagement are corroborated by De Oliveira et al. (2019), who note that using this individualized learning system that provides students with engaging digital instructional content increased students' engagement and motivation to learn. Alkhalisi's (2019) definition of Alef as an interactive shareware software that customizes the content delivered to each student based on their needs, feedback, and recognized strengths and weaknesses also supports the study's findings. Further evidence also supports Alef's claim of providing rapid feedback and enabling teachers and parents to access analytical data and information to gauge students' progress.

According to Eaton et al. (2018), another benefit of employing Alef is its capacity to promote differentiation in learning as well as individual learning. Teachers can concentrate their efforts on differentiating learning for distinct groups of pupils due to the immediate feedback they receive on the progress of their students. Thus, Eaton et al. (2018) corroborate

the key benefits of Alef, such as student autonomy in learning and frequent teacher feedback on student progress. A learning management system, according to Adzharuddin and Ling (2013), is a necessary tool for university students since it allows them to stay on top of their schoolwork and receive fast alerts about their daily tasks. As a result, teachers find it simpler to communicate with their students outside of the classroom and can quickly inform them via the system about problems with their homework. This study's findings are also supported by Alahmari and Kyei-Blankson (2016), who argue that learning management systems allow teachers to provide feedback to students, guiding them toward better learning outcomes.

A contradiction was found with Alhosani (2019), who argues that utilizing a computer is one of the hurdles of the Alef platform. However, the current study saw computer use as a motivator for learning throughout the Alef program. Parents believed that the students' motivation came from using technology through Alef. Similarly, Alahmari and Kyei-Blankson (2016) noted the issue of internet access as a major challenge, which was not confirmed in this study due to the fact that the UAE is a technologically advanced country with a high internet penetration rate.

The findings of the current study uncovered several crucial challenges that have an impact on student learning when using Alef. Parents suggested that certain lesson questions were too difficult. The difficulty of these questions may hinder students' learning and cause them to abandon their attempts to solve them. Parents also mentioned the issue of students being under tremendous time and pressure constraints as a significant challenge. Through the Alef program, students receive tasks from multiple academic disciplines. This results in students spending long hours in front of the computer. In addition, the time given may not be sufficient, which may lead to feelings of frustration and aversion for the students. Therefore, a solution must be found that provides a balance in the use of the program while ensuring that it retains its specific learning-supporting characteristics.

Another contradiction was found with Akkary (2014), who indicated that the presence of numerous cultural groups in the United Arab Emirates presents a language hurdle when using the Alef program. Despite this, the current study's findings did not definitively establish that language is a barrier to using the program. This could be because English is so widely used in Emirati culture. Perhaps this is because all public schools now use English as their primary language of instruction. However, parents in this study favored the addition of an integrated translation system to the application, particularly for translating scientific terminology. Students and parents alike will benefit from this translation tool.

Students may become disinterested in using Alef due to its perceived fixed, monotonous interface; therefore, it is important to pay attention to the program's outward appearance and make it appealing to students by regularly updating its design. This finding is corroborated by Gee (2022) and Kumar et al. (2020), who confirmed the issue of fixed and monotonous interfaces, which should be managed according to students' preferences. Arranging students' names in a random order is also not ideal; alphabetical order is preferred so that parents can find their children's names more quickly. All parents surveyed by Abdallah (2018) agreed that the program needs to improve the process of parents' access to their children's progress (Abdallah 2018).

## 6. Conclusions and Recommendations

In conclusion, Alef was found to support students' learning by enhancing their independence and allowing them to engage in deeper learning. The system was found to be effective for both learning and teaching due to its interactive capabilities, although some constraints were found. Suggested recommendations based on the findings of the study and merit considerations by Alef program administrators will now be presented. One of these suggestions is improving the material covered in Alef lessons, for example, cutting down on some activities and coordinating between subjects. This recommendation follows that of Bieliková et al. (2014), who stated that it is possible to distinguish between different types of learning objects (explanations, exercises, and questions) and that this



allows recommendations to be more specific for particular learning objects. Bieliková et al. (2014) based their study on Alef logs as well as external sources of crucial data on students, such as manual evaluations outside of Alef or personality factors, which serve as important sources of advice. Conversely, it is essential to increase the questions on Alef that support end-of-year examinations; this requires that the Alef questions have the same frequency as the end-of-year examinations. Despite the range and variety of Alef's questions, the high-level exit questions may be frustrating for some students at times. In addition, the questions must be focused on what is covered in the textbook. For instance, social studies is a subject based mainly on acquiring skills; the final exam targets certain skills, and the student needs training in these skills. In contrast, Alef is dedicated solely to the dissemination of knowledge and does not give attention to these skills.

*Recommendations for Future Research*

It is important to note that the findings of this study were constrained by the extremely limited literature that could be found to serve as a background and contrast for discussion. This can be managed by conducting more studies with larger sample sizes. Polling a sizable number of parents may reveal new advantages, difficulties, and suggestions that could expand the body of research and assist in improving the Alef system. It is also recommended that comparative studies should be conducted to assess the performance of students in different Emirates after utilizing Alef to evaluate the adoption and outcomes of the system in all the Emirates of the country.

**Author Contributions:** Conceptualization, N.E. and N.A.; methodology, O.A.K.; software, O.A.K.; validation, N.A.; formal analysis, N.E.; investigation, N.A.; resources, O.A.K.; data curation, N.E.; writing—original draft preparation, N.E.; writing—review and editing, N.A.; visualization, O.A.K.; supervision, N.A.; project administration, O.A.K.; funding acquisition, N.A. All authors have read and agreed to the published version of the manuscript.

**Funding:** This research received no external funding and the APC was funded by authors.

**Institutional Review Board Statement:** The study was conducted in accordance with the Declaration of Helsinki, and approved by the Research Ethics Committee of Emirates College for Advanced Education (IRB Reference Number: SP-10-2021 and date of approval is 18 February 2021).

**Informed Consent Statement:** Informed consent was obtained from all subjects involved in the study.

**Data Availability Statement:** The data presented in this study are available on request from the corresponding author. The data are not publicly available due to the privacy of the interviewed parents.

**Conflicts of Interest:** The authors declare no conflict of interest.

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
