# Peer review of "Adaptive Learning Framework (Alef) in UAE Public Schools from the Parents’ Perspective"

_socsci, doi:10.3390/socsci12050297_

Round 1

Reviewer 2 Report

1. The script has been well written.

2. The rationality described in the introduction is quite good.

3. The research method is good, clear and rational.

4. The text can be accepted scientifically and empirically.

5. The research results have good novelty and state of the art.

6. Drawings/tables/schematics are appropriate, and made based on accurate data.

7. The author presents valid and reliable data.

8. The text is easy to interpret and understand.

9. The data has been properly interpreted.

10. Conclusions have been presented in a manner consistent with the evidence and arguments presented.

11. The study has been written in a clear, comprehensive, and relevant to the field.

Reviewer 3 Report

The article is certainly interesting, as it provides an opportunity to look at online learning through the eyes of parents. The problems of online learning in the article are considered on the example of only one adaptive system "ALEF", but it is obvious that these problems are of a general nature.

It seems to me that the authors pay too much attention to the description of the ALEF system itself. For example, the description of the concept of "artificial intelligence" is found in the article in the sections "Introduction" and "Theoretical basis". For the main content of the article, this concept (artificial intelligence) does not play any role. By the way, I note that the "Theoretical Basis" section is very similar to an advertisement for the ALEF system. On the other hand, the other sections are very brief. For example, nowhere is there a list of questions that were offered to survey participants. Table 1 looks somewhat strange, since all the participants in the survey are women wuth degree bachelors.

In the "Results" section, the boundary between the text of the authors of the article and the answers of the parents should be made clearer. Perhaps it makes sense to end each part of this section with conclusions. Now all the conclusions are collected in the "Discussion" part.

Conclusion: the article needs to be seriously revised.

Reviewer 4 Report

Actually, this paper is more of an evaluation study than a research study. The paper is still one that should be published, but it would be more credible if the authors clearly indicate that evaluation of the Alef was a goal.